# Hepatitis C Virus Proteins Core and NS5A Are Highly Sensitive to Oxidative Stress-Induced Degradation after eIF2α/ATF4 Pathway Activation

**DOI:** 10.3390/v12040425

**Published:** 2020-04-09

**Authors:** W. Alfredo Ríos-Ocampo, María-Cristina Navas, Manon Buist-Homan, Klaas Nico Faber, Toos Daemen, Han Moshage

**Affiliations:** 1Department of Gastroenterology and Hepatology, University Medical Center Groningen, University of Groningen, 9713 GZ Groningen, The Netherlands; m.buist-homan@umcg.nl (M.B.-H.); k.n.faber@umcg.nl (K.N.F.); a.j.moshage@umcg.nl (H.M.); 2Department of Medical Microbiology, University Medical Center Groningen, University of Groningen, 9713 GZ Groningen, The Netherlands; c.a.h.h.daemen@umcg.nl; 3Gastrohepatology Group, Medicine School, University of Antioquia, Medellin 050010, Colombia; maria.navas@udea.edu.co

**Keywords:** hepatitis C virus, Core, NS5A, oxidative stress, autophagy-adaptor proteins, eIF2a/ATF4 pathway

## Abstract

Hepatitis C virus (HCV) infection is accompanied by increased oxidative stress and endoplasmic reticulum stress as a consequence of viral replication, production of viral proteins, and pro-inflammatory signals. To overcome the cellular stress, hepatocytes have developed several adaptive mechanisms like anti-oxidant response, activation of Unfolded Protein Response and autophagy to achieve cell survival. These adaptive mechanisms could both improve or inhibit viral replication, however, little is known in this regard. In this study, we investigate the mechanisms by which hepatocyte-like (Huh7) cells adapt to cellular stress in the context of HCV protein overexpression and oxidative stress. Huh7 cells stably expressing individual HCV (Core, NS3/4A and NS5A) proteins were treated with the superoxide anion donor menadione to induce oxidative stress. Production of reactive oxygen species and activation of caspase 3 were quantified. The activation of the eIF2α/ATF4 pathway and changes in the steady state levels of the autophagy-related proteins LC3 and p62 were determined either by quantitative polymerase chain reaction (qPCR) or Western blotting. Huh7 cells expressing Core or NS5A demonstrated reduced oxidative stress and apoptosis. In addition, phosphorylation of eIF2α and increased ATF4 and CHOP expression was observed with subsequent HCV Core and NS5A protein degradation. In line with these results, in liver biopsies from patients with hepatitis C, the expression of ATF4 and CHOP was confirmed. HCV Core and NS5A protein degradation was reversed by antioxidant treatment or silencing of the autophagy adaptor protein p62. We demonstrated that hepatocyte-like cells expressing HCV proteins and additionally exposed to oxidative stress adapt to cellular stress through eIF2a/ATF4 activation and selective degradation of HCV pro-oxidant proteins Core and NS5A. This selective degradation is dependent on p62 and results in increased resistance to apoptotic cell death induced by oxidative stress. This mechanism may provide a new key for the study of HCV pathology and lead to novel clinically applicable therapeutic interventions.

## 1. Introduction

Hepatitis C virus (HCV) is a member of the *Flaviviridae* family and was identified in 1989 as the infectious agent of non-A, non-B hepatitis. Currently, HCV is the leading cause of end-stage liver disease as a result of cirrhosis and/or hepatocellular carcinoma (HCC). An estimated 71 million people are chronically infected and approximately 400,000 associated deaths occur each year worldwide [1,2]. Although safe, tolerable and curative therapies for HCV infection have emerged in recent years, the prevention, clinical management and access to treatment remain important determinants in the control of HCV infection. Despite the recent therapeutic advances, HCV pathophysiology is still not entirely elucidated justifying continued research in this field [3].

HCV-infected hepatocytes are exposed to several stressors that may affect their function and viability. These stressors include viral replication and viral protein production within hepatocytes, as well as the inflammatory response of the host. It is known that HCV infection leads to increased oxidative stress in the liver and in particular in the hepatocytes [4]. Since HCV replication and viral protein production are closely linked to the endoplasmic reticulum (ER), both ER stress and oxidative stress occur and contribute to the progression of chronic HCV-related liver disease; therefore, hepatocytes should adapt to injury insult [5,6,7,8,9]. However, little is known about the consequences of this adaptation upon HCV infection.

HCV contains a positive sense single-stranded RNA (ssRNA+) genome that encodes for a polyprotein of approximately 3100 amino acids, depending on the genotype, that is cleaved co- and post-translationally by cellular and viral proteases to produce 10 viral proteins with various structural and biochemical functions (Core, E1, E2, p7, NS2, NS3, NS4A, NS4B, NS5A and NS5B) [10]. The role of HCV proteins in the generation of oxidative stress and ER stress has been demonstrated and Core and the non-structural proteins NS3/4A and NS5A are the most potent inducers [11,12,13].

In mammalian cells, different signaling pathways have evolved to mediate the cellular stress response. One of the most conserved regulatory events activated in response to stress is the phosphorylation of the α subunit of eukaryotic translation Initiation Factor 2 (eIF2α) at serine 51 and subsequent ATF4 (Activation Transcription Factor 4) activation [14]. It has been demonstrated that activation of the eIF2α/ATF4 pathway directs an autophagy gene transcriptional program to overcome cellular stress. Furthermore, the transcription factors ATF4 and CCAAT/Enhancer-Binding Protein Homologous Protein (CHOP) are involved in the transcriptional activation of other autophagy genes, including p62/SQSTM1 (Sequestosome 1) [hereafter referred to as p62] [15].

Autophagy can also be induced via activation of the Unfolded Protein Response (UPR) through phosphorylation of the ER stress sensors: Protein Kinase R (PKR)-like endoplasmic reticulum kinase (PERK) and Inositol-Requiring protein 1 (IRE1, or Endoplasmic reticulum to nucleus signaling 1, human homologue). Both sensors can induce and activate Beclin-1, as well as the expression of autophagy-related genes (ATGs), *ATG5* and *ATG12* [16,17]. Additionally, PERK phosphorylation is known to couple distinct upstream stress signals to eIF2α/ATF4 pathway activation and further autophagy gene expression [15]. In Huh7 hepatoma cells, infection with HCV leads to induction of the UPR and subsequently to an inhibition of the phosphatidylinositol 3-kinase (PI3K)/AKT/mammalian Target of the Rapamycin (mTOR) signaling pathway, resulting in induction of autophagy [16]. HCV-induced autophagy can also be mediated via an UPR-independent mechanism, since silencing of the UPR-sensor IRE1 in HCV-infected Huh7.5 cells did not affect HCV replication or the induction of autophagy [18].

In a previous study, we established a model in Huh7 cells, as well as in primary rat hepatocytes, to investigate the adaptive responses activated under HCV protein expression and external cellular oxidative stress induction. HCV Core or NS3/4A were transiently expressed in hepatocytes and subsequently treated with menadione, a superoxide anion donor. We observed that under induction of external oxidative stress and HCV protein expression, both pro-oxidants, hepatocytes adapt to oxidative stress via a reduction in reactive oxygen species (ROS) production as well as reduction of oxidative stress-induced apoptosis [19]. In addition, we observed an increased degradation of the HCV Core protein together with the autophagy adaptor protein p62 in hepatocytes resistant to oxidative stress [19].

In the present study, we investigated the adaptive response to stress in cells overexpressing HCV proteins. Autophagy-related proteins and changes in their steady state protein levels were investigated and related to protection against cell death in the context of HCV protein overexpression, oxidative stress and ER stress. We observed that selective degradation of HCV Core and NS5A proteins via activation of the eIF2α/ATF4 pathway plays an important role in the adaptive response of hepatocytes to stress via suppression of pro-oxidant agents.

## 2. Material and Methods

### 2.1. Cell Lines and Culture

Huh7 cells were maintained in Dulbecco’s Modified Eagle Medium (1X) + GlutaMAX^TM^- I (DMEM; Gibco, Landsmeer, The Netherlands) supplemented with 10% fetal bovine serum (FBS, Gibco, Landsmeer, The Netherlands) and 1% penicillin-streptomycin (Gibco, Landsmeer, The Netherlands) at 37 °C and 5% CO_2_. The stable cell lines Huh7_puro (3 µg/mL puromycin) (containing the empty vector), Huh7_Core_Jc1_bla (10 µg/mL blasticidin), Huh7_NS3/4A_Con1 (0.75 µg/mL G418) and Huh7_NS5A_JFH1_puro (3 µg/mL puromycin) were generated and kindly provided by Prof. Dr. Ralf Bartenschlager from the University of Heidelberg, Germany [20].

### 2.2. Reagents and Treatments

Cells (3.0 × 10^5^) were grown until 80% confluence in 6-well plates 24 h (h) prior to treatment. Cells were treated for 6 h with 50 µmol/L menadione (Sigma, Zwijndrecht, The Netherlands) or 5 mmol/L hydrogen peroxide (H_2_O_2_) (Sigma, Zwijndrecht, The Netherlands) to induce oxidative stress. Subsequently, cells were harvested and viability was determined by trypan blue exclusion staining. For the kinetic assays cells were collected after menadione treatment every hour for 6 h. In control experiments, 5 mmol/L *N*-acetyl-l-cysteine (NAC, Sigma, Zwijndrecht, The Netherlands) was added 30 min prior to menadione treatment. In some experiments 50 µmol/L chloroquine (CQ) diphosphate salt (Sigma, Zwijndrecht, The Netherlands) was used to inhibit autophagic flux and 100 nmol/L bafilomycin A1 (Sigma, Zwijndrecht, The Netherlands) or a mixture of 20 mmol/L ammoniumchloride (NH_4_Cl)/100 µmol/L leupeptin/100 µmol/L pepstatin was used to inhibit the lysosomal degradation pathway. For proteasome inhibition 10 µmol/L MG132 (Sigma, Zwijndrecht, The Netherlands) was added 3 h prior to menadione treatment. Experiments were conducted in duplicate and the results are expressed as the average of three independent experiments.

### 2.3. Tissue Samples

Liver tissue samples from patients submitted for liver transplantation were obtained from the tissue bank of the Gastrohepatology Group, University of Antioquia, Colombia. The samples were selected according to etiology and used for RNA isolation and quantitative polymerase chain reaction (qPCR) analysis. The liver tissues were from patients with cirrhosis associated to HCV infection (6 patient samples), HCV-associated hepatocellular carcinoma (HCC) (7 patient samples), hepatitis B virus (HBV)-associated cirrhosis (4 patient samples) and non-viral liver disease (4 patient samples). The demographic and clinical characteristics of the samples are presented in Table 1. For total RNA isolation 100 mg of tissue was processed using TriZOL reagent (Invitrogen, Landsmeer, The Netherlands) following the manufacturer’s instructions. Total RNA (2.5 µg) was used for reverse transcription (RT). Complementary DNA (cDNA) was diluted 20 X in nuclease free water and stored at −20 °C until use.

### 2.4. Cell Culture, RNA Isolation and Quantitative Polymerase Chain Reaction (qPCR)

After treatment, Huh7 cells stably expressing the empty vector, Core, NS3/4A and NS5A were harvested on ice and washed three times with ice-cold 1X Hank’s Balanced Salt Solution (HBSS) with Ca^2+^ and Mg^2+^ (Gibco, Landsmeer, The Netherlands). TRI reagent (Sigma, Zwijndrecht, The Netherlands) was added to the cells for total RNA isolation according to the manufacturer’s instructions. Total RNA (2.5 µg) was used for RT in 1X RT buffer (500 mmol/L Tris-HCl -pH 8.3-; 500 mmol/L KCl; 30 mmol/L MgCl_2_; and 50 mmol/L DTT), 1 mmol/L deoxynucleotides triphosphate (dNTPs, Sigma, Zwijndrecht, The Netherlands), 10 ng/µL random nanomers (Sigma, Zwijndrecht, The Netherlands), 0.6 U/µL RNaseOUT^TM^ (Invitrogen, Landsmeer, The Netherlands) and 4 U/µL Moloney-Murine Leukemia Virus (M-MLV) reverse transcriptase (Invitrogen, Landsmeer, The Netherlands) in a final volume of 50 µL. cDNA was diluted 20 X in nuclease free water and qPCR was carried out in a StepOnePlus™ (96-well) PCR System (Applied Biosystems, Landsmeer, The Netherlands) using TaqMan probes; the sequences of the probes and primers are described in Appendix A. For qPCR, 2X reaction buffer (dNTPs, HotGoldStar DNA polymerase, 5 mmol/L MgCl_2_) (Eurogentech, Maastricht, The Netherlands), 5 µmol/L fluorogenic probe and 50 µmol/L of primers sense and antisense (Invitrogen, Landsmeer, The Netherlands) were used. mRNA levels were normalized to 18S gene expression and compared between groups [21]. The experiments were performed in duplicate and presented as the average of three independent experiments.

### 2.5. Determination of Cellular Oxidative Stress

Total cytoplasmic ROS was quantified using the fluorogenic probe CellROX^®^ Deep Red (Invitrogen, Landsmeer, The Netherlands) following the manufacturer‘s instructions. After induction of oxidative stress, 5 µmol/L of CellROX reagent was added to the cells and cells were subsequently incubated at 37 °C and 5% CO_2_ for 30 min. Media was removed and cells were washed three times with 1X HBSS with Ca^2+^, Mg^2+^ (Gibco, Landsmeer, The Netherlands) and subsequently harvested using 1X trypsin (Gibco, Landsmeer, The Netherlands) and analyzed by flow cytometry using a BD FACSVerse system and a 635nm laser. Three independent experiments were carried out and the results are expressed as an average.

### 2.6. Caspase 3 Activity Determination and Flow Cytometry

After treatment, cells were scraped on ice and lysed by three cycles of freezing (liquid nitrogen) and thawing (37 °C) in lysis buffer (25 mmol/L HEPES, 150 mmol/L KAc, 2 mmol/L EDTA, 0.1% NP-40) supplemented with protease and phosphatase inhibitors (10 mmol/L NaF, 50 mmol/L PMSF, 1 µg/µL of α-protenin/pepstatin/leupeptin and 1 mmol/L DTT) followed by centrifugation for 10 min at 12,000 rpm. For the caspase 3 activity assay [22], 30 µg of protein was mixed with the synthetic fluorogenic caspase 3 substrate, Ac-DEVD-AMC and the release of fluorogenic AMC was quantified in a spectrofluorometer at an excitation wavelength of 380 nm and emission wavelength of 430 nm. The arbitrary units of fluorescence (AUF) from three independent experiments were used to depict the results. Multiparametric apoptosis assay by flow cytometry was performed using MitoProbe^TM^ DilC(5) assay kit in combination with propidium iodide (PI) staining following the manufacturer’s protocol (ThermoFisher Scientific, Landsmeer, The Netherlands). Cells were harvested after menadione treatment (50 µmol/L) using trypsin. Then, 10 µM DilC(5) and 100 µg/mL PI were added to the cells.

### 2.7. Transfection of siRNA

For silencing of p62, 4 × 10^4^ cells were seeded in 12-well plates pre-treated with 1.5 µL Lipofectamine 3000 (Invitrogen, Landsmeer, The Netherlands) and 50 μmol/L esiRNA human p62/SQSTM 1 (esiRNA1) (Sigma, Zwijndrecht, The Netherlands, Cat. #EHU027651) or scrambled siRNA (esiRNA Egfp Cat. #EHUEGFP-20UG, Sigma, Zwijndrecht, The Netherlands) as a control. The Lipofectamine 3000 and the esiRNAs were prepared in 75 μL OPTI-MEM^TM^ I (1X) reduced serum medium (Gibco, Landsmeer, The Netherlands) following the manufacturer’s instructions then wells were cover with the mixture (reverse transfection). After cells were added, media was completed to 750 μL final volume per well. 12 h post-transfection, media were replaced and a second transfection round was performed using the same amounts of Lipofectamine 3000 and esiRNA for 12 h. Subsequently, cells were treated with 50 μmol/L menadione for 6 h. After treatment cells were scraped and lysed by freezing and thawing cycles in lysis buffer containing protease and phosphatase inhibitors as described above, followed by centrifugation for 10 min at 12,000 rpm. Supernatant was collected and stored at −20 °C until use. Three independent experiments were performed and the results are expressed as means ±S.D.

### 2.8. Western Blotting

Cell lysates (20 μg) were resolved on Mini-PROTEAN^®^ TGX Stain-Free^TM^ Precast Gels (BioRad, Veenendaal, The Netherlands) and semi-dry blotting transfer was performed using Trans-Blot Turbo Midi Nitrocellulose Membrane with Trans-Blot Turbo System (BioRad, Veenendaal, The Netherlands). To confirm the electrophoretic transfer, Ponceau S 0.1% *w*/*v* (Sigma, Zwijndrecht, The Netherlands) staining was used. The monoclonal antibodies human anti-HCV Core B12-F8 (kindly provided by prof. Dr. Mondelli, University of Pavia, Italy) [23], mouse anti-HCV NS3/4A (8 G-2) (Abcam, Cambridge, UK) and mouse anti-HCV NS5A (9E10) (kindly provided by prof. Dr. Charles M. Rice), were used at a dilution of 1:1000 and mouse anti-glyceraldehyde 3-phosphate dehydrogenase (GAPDH) (Calbiochem, Amsterdam, The Netherlands) at a dilution of 1:10,000. ER stress markers were also determined using the polyclonal rabbit antibodies anti-peIF2α (Cell Signaling, Leiden, The Netherlands), anti-eIF2α (total) (Cell Signaling, Leiden, The Netherlands), anti-ATF4 (Cell Signaling, Leiden, The Netherlands) and anti-glucose-regulated orotein of 78kDa (GRP78) (Cell Signaling, Leiden, The Netherlands) at 1:1000 dilution. Polyclonal rabbit anti-Microtubule Associated Protein 1 Light Chain 3 Beta (LC3B) (Cell Signaling, Leiden, The Netherlands) and anti-p62 (Cell Signaling, Leiden, The Netherlands) were also used at 1:1000 dilution. For detection of ubiquitinated proteins mouse anti-α-Ubiquitin (1/1000) (Hycult, Biotech, Uden, The Netherlands) was used. Secondary horseradish peroxidase (HRP)-bound antibodies were used. The blots were analyzed by chemiluminescence in a ChemiDoc XRS system (Bio-Rad, Veenendaal, The Netherlands). Protein band intensities were quantified by ImageLab software (BioRad, Veenendaal, The Netherlands).

### 2.9. Statistical Analysis

All experiments were performed at least three times and the mean ± standard deviation (s.d.) is depicted. The Graphpad Prism 5 software (GraphPad Software) was used and comparisons were evaluated by unpaired, two-tailed *t*-test. For the group analysis two tails Anova and Bonferroni post-test were performed. A *p* value of <0.05 was considered statistically significant.

## 3. Results

### 3.1. Reactive Oxygen Species (ROS) Generation Is Attenuated in Huh7 Cells Expressing NS3/4A and NS5A

We first investigated the generation of ROS in Huh7 cells, stably expressing either Core, NS3/4A or NS5A (Appendix A) with and without menadione treatment as second external stressor. As expected, Huh7 cells transfected with empty vector showed an increase in ROS production after menadione treatment and this effect was suppressed by NAC pre-treatment (Figure 1A). A similar response was observed for Huh7 cells expressing Core (Figure 1B). Remarkably, the ROS generation in response to menadione in Huh7 cells stably expressing NS3/4A (Figure 1C) or NS5A (Figure 1D) was much less prominent compared to empty vector and Core expressing cells, suggesting that after external oxidative stress the HCV proteins NS3/4A and NS5A attenuate menadione-induced ROS generation and therefore cellular oxidative stress. Importantly, basal levels of ROS production were similar in all 4 cell lines.

### 3.2. Huh7 Cells Expressing Hepatitis C Virus (HCV) Core and NS5A Are Protected against Apoptosis Induced by Oxidative Stress

We next investigated the effect of expression of HCV proteins on susceptibility to oxidative stress-induced toxicity. As shown in Figure 2A, menadione treatment alone induced caspase 3 activation in Huh7 cells expressing the empty vector and this effect could be completely reversed by the antioxidant NAC (Figure 2A). Huh7 cells expressing Core or NS5A (Figure 2B,D), but not NS3/4A (Figure 2C), were completely protected against menadione-induced caspase 3 activation (Figure 2B,D). Interestingly, NAC treatment abolished the protective effect observed in Huh7 cells expressing HCV Core or NS5A on menadione-induced apoptosis (Figure 2B,D). The number of pre-apoptotic cells measured by flow cytometry in Huh7 cells expressing the empty vector and Huh7 cells expressing HCV Core and NS5A after exposure to oxidative stress was reduced in cells expressing viral proteins (Appendix A). These results suggest that after external oxidative stress, hepatocytes expressing Core or NS5A viral proteins activate an adaptive response to attenuate caspase 3 activation and thereby the apoptotic effect of oxidative stress.

### 3.3. The eIF2α/ATF4 Pathway Is Involved in the Adaptive Response against External Oxidative Stress-Induced Damage

To elucidate the mechanism behind the protective effect of Core and NS5A against menadione-induced apoptosis, first, the transcriptional regulation of genes encoding antioxidant enzymes (copper zinc superoxide dismutase [CuZnSOD, *SOD1*] and catalase [*CAT*]) were determined. No regulation of antioxidants genes was observed (Appendix A), suggesting that other mechanisms are involved in the adaptive resistance to oxidative stress.

An important pathway in the adaptation to cellular oxidative stress is the phosphorylation of eIF2α and subsequent induction of ATF4 and CHOP expression [14]. The eIF2α/ATF4 pathway has been associated with the induction of autophagy after cellular stress as an essential mechanism of survival [15]. Therefore, we investigated the eIF2α/ATF4 pathway in our model. Menadione robustly induced phosphorylation on serine 51 of eIF2α of Huh7 cells expressing the empty vector, Core, NS3/4A or NS5A, which was completely abolished by antioxidant treatment with NAC (Figure 3A). Remarkably, mRNA levels of *ATF4* were not significantly increased after menadione treatment in any of the four Huh7 cell lines probably because qPCR measurements were performed at a late time point, therefore, the transcriptional effect was lost (Appendix A). However, at the protein level, increased ATF4 protein expression was observed after menadione treatment in Huh7 cells expressing the empty vector and all the HCV viral proteins and this increase was reversed by the antioxidant NAC (Figure 3B). The expression of CHOP (*DDIT3*) is required for the transcription of a set of autophagy genes after activation of the eIF2α/ATF4 pathway. *DDIT3* mRNA levels were significantly increased after exposure to oxidative stress in Huh7 cells expressing the empty vector and its expression was diminished upon NAC treatment. However, in cells expressing Core and NS5A even after treatment with NAC, *DDIT3* mRNA was still increased indicating the importance of HCV Core or NS5A in *DDIT3* induction (Figure 3C). The expression of the ER stress marker GRP78 (*HSPA5*) was also analyzed to determine whether the eIF2α/ATF4 pathway was activated as a consequence of ER stress. We did not observe any regulation of *HSPA5* mRNA or GRP78 protein levels in any of the cell lines and by any of the treatments (Figure 3A,D). This result suggests that the eIF2α/ATF4 pathway was not activated in response to ER stress, but as an adaptive response to attenuate external oxidative stress.

### 3.4. Expression of ATF4 and DDIT3 Is Induced in HCV-Related Cirrhosis

To confirm our in vitro data in clinical samples, we determined the mRNA expression of transcription factors *ATF4* and *DDIT3* in liver biopsies from patients with end-stage liver diseases (Table 1). Both *ATF4* and *DDIT3* mRNA expression were significantly increased in samples of patients with HCV-related cirrhosis, where HCV viral load is fluctuating compared to the tumor tissue of patients with HCV-associated HCC where HCV viral load decrease [24] (Figure 4A,B). The increased expression of *ATF4* and *DDIT3* was specific for HCV-related cirrhosis since their expression was not induced in liver samples of patients with non-HCV end-stage liver disease as HBV-associated cirrhosis (Figure 4A,B). These results underscore the relevance of our in vitro findings for more clinical conditions.

### 3.5. HCV Core and NS5A are Preferentially Degraded after Menadione Treatment: Involvement of Autophagy Adaptor Proteins

Activation of the eIF2α/ATF4 pathway in response to stress can induce autophagy as a survival mechanism [15]. Therefore, we investigated changes in the steady state protein levels of LC3 and the autophagy adaptor protein p62 and determined whether these changes correlated with increased resistance to external oxidative stress. In cells expressing the empty vector, or any of the HCV proteins, the steady state levels of LC3-I appear to be increased whereas LC3-II diminished significantly after menadione treatment which was recovered by the antioxidant NAC. A similar pattern was observed with the expression of p62 (Figure 5A–D). Thus, after menadione treatment, both p62 and LC3-II were virtually absent in all Huh7 cell lines, which suggest a relative modulation of the steady state of the autophagy markers after external oxidative stress induction.

Menadione treatment also induced the loss of HCV Core and NS5A in the Huh7 cells. This degradation was reversed by the antioxidant NAC (Figure 5B,D). This result was not observed for NS3/4A (Figure 5C) which was apparently upregulated, suggesting that HCV Core and NS5A degradation is a selective process. The degradation of Core and NS5A viral proteins occurred simultaneously with a decrease in the levels of p62 and LC3-II, which may account for their degradation, suggesting that autophagy adaptor proteins may play an important role in the selective elimination of Core and NS5A after external oxidative stress induction.

To confirm that oxidative stress affects the protein levels of HCV Core and NS5A, we used H_2_O_2_ as an alternative source of external oxidative stress and indeed we obtained similar results, a pronounced loss of Core which was reversed by the anti-oxidant NAC (Appendix A). NS3/4A levels did not change in the presence of H_2_O_2_ (Appendix A). In contrast, the NS5A protein level diminished after H_2_O_2_ treatment, which indicates increased susceptibility of NS5A to degradation after external oxidative stress. This particular effect was not reversed by NAC treatment (Appendix A). These data suggest that HCV proteins Core and NS5A are highly sensitive to oxidative stress-induced degradation.

### 3.6. The Ubiquitin-Proteasome Pathway Is Not Involved in HCV Core and NS5A Degradation

The ubiquitin-proteasome pathway selectively degrades a wide range of protein substrates employing a ubiquitin conjugation cascade. We investigated whether the proteasome pathway is involved in HCV Core and NS5A degradation after external oxidative stress. Huh7 cells were treated with the proteasome inhibitor MG132 and accumulation of ubiquitinated proteins was observed after 2 h (Figure 6A). As already shown in Figure 5, menadione treatment reduced protein levels of Core and NS5A, but not NS3/4A (Figure 6B–D). Pre-treatment of Huh7 cells expressing Core or NS5A with MG132 did not prevent the degradation of Core and NS5A in Huh7 cells exposed to menadione (Figure 6B,C). Huh7 cells expressing HCV NS3/4A were used as a control and MG132 treatment did not affect protein levels of NS3/4A (Figure 6D). These results suggest that the proteasome pathway is not involved in the degradation of HCV Core and NS5A proteins.

### 3.7. Menadione-Induced NS5A Degradation Involves the Autophagy-Related Proteins LC3 and p62

Autophagic flux was blocked using chloroquine (CQ) to determine whether lysosomal degradation of HCV proteins occurs after autophagolysosome formation. The experiments were performed using Huh7 cells expressing NS5A since in these cells we observed a complete degradation of the viral protein after menadione treatment and strong recovery after NAC treatment (Figure 5D). As shown in Figure 7A, treatment of Huh7 cells expressing NS5A with only CQ showed accumulation of LC3-II and p62 which indicates inhibition of the autophagy flux (Figure 7A). Consistent with previous results, menadione treatment reduced the levels of LC3-II, p62 and NS5A. When autophagic flux was blocked in cells exposed to menadione (+/+), no recovery of NS5A or p62 protein expression was observed. Since we were expecting at least a recovery in p62 expression after autophagy flux inhibition we conclude that p62 is involved in NS5A degradation via a degradative pathway not involving autophagolysosome formation (Figure 7A). We also explored the effect of other inhibitors of lysosomal degradation, Bafilomycin A1 and a mix of protease inhibitors (NH_4_Cl/leupeptin/pepstatin). After cells were treated with Bafilomycin A1 and exposed to menadione (+/+), NS5A protein levels were partially restored. Although full restoration of NS5A protein levels was not observed, the recovery of the NS5A protein was statistically significant as demonstrated by densitometry analysis (Figure 7B). Similar results were observed using NH_4_Cl/leupeptin/pepstatin as the inhibitor of the lysosomal pathway (Figure 7C). The above results suggest that inhibition of lysosomal degradation partially restored the NS5A protein level.

### 3.8. A Role for the Autophagic Adaptor Protein p62 in the Degradation of Core and NS5A

To further analyze the role of p62 in HCV Core and NS5A protein degradation, we performed a time course of the degradation of HCV Core, NS5A and p62 in Huh7 cells (Figure 8A,B). Protein levels of HCV Core and NS5A remained relatively stable during the first 2 h of menadione treatment, after which their levels rapidly dropped in the following hour(s) (Figure 8A,B).

Remarkably, a virtually identical response of p62 to menadione was observed (Figure 8A,B). Since degradation of p62 parallels the elimination of HCV Core and NS5A, and since p62 functions as an adaptor protein, we further investigated the relation between p62 and the viral proteins by selective p62 silencing. Expression of p62 was suppressed in Huh7 cells using siRNA technology and Huh7 cells transfected with a random control siRNA served as controls (Figure 8C and D). As expected, degradation of Core, NS5A and p62 was observed after exposure to menadione, (Figure 8C,D). When p62 was silenced prior to menadione treatment of Huh7 cells, the expression of Core and NS5A were significantly recovered (Figure 8C,D). These results confirm the role of p62 as an adaptor protein for selective degradation of HCV Core and NS5A after external oxidative stress.

## 4. Discussion

The goal of the present study was to investigate the adaptive response of hepatocyte-like cells to multiple stressors including oxidative stress and overexpression of pro-oxidant HCV proteins. Our findings can have relevance for other liver diseases accompanied by increased oxidative stress, e.g., non-alcoholic steatohepatitis (NASH) and alcoholic steatohepatitis (ASH). We found that Huh7 cells expressing HCV Core and NS5A resist the deleterious effects from additional oxidative stress via selective degradation of these viral proteins involving autophagy adaptor proteins such as p62 and LC3. The degradation occurs in response to activation of the eIF2α/ATF4 pathway and suggest elimination of harmful viral proteins that could increase the cytotoxic effects from oxidative stress (Figure 9).

Infection of cells by viruses constitutes an important stress to the host cells. Cells must support viral replication and synthesis and shedding of newly synthesized virions. In addition, in most viral infections, cells are also exposed to immune and/or inflammatory response to the viral infection. Therefore, cells have to adapt to multiple stressors in order to survive and thus also sustain the viral replication cycle. HCV infection of hepatocytes is an example in which cells are subjected to multiple stressors: viral protein synthesis may lead to ER stress and HCV infection is also accompanied by oxidative stress [4,25]. In a previous study, we observed increased resistance to menadione-induced oxidative stress using Huh7 cells and primary rat hepatocytes transiently expressing HCV Core or NS3/4A proteins. Both mitochondrial ROS production and menadione-induced apoptosis were significantly decreased together with reduced levels of the ER stress markers GRP78 and sXBP1 in Huh7 cells expressing HCV Core and NS3/4A proteins. This increased resistance was accompanied by increased degradation of the HCV Core protein in these cells, suggesting that selective degradation of one stressor may be in part responsible for this increased resistance [19]. In the present study, using stably transfected Huh7 cells, we confirm increased resistance to oxidative stress and the selective degradation of HCV proteins Core and NS5A, as well as the activation of the stress pathway eIF2α/ATF4. Although the infectious HCV culture system (HCVcc) was established in 2005 and is widely used in the field for the study of the cellular responses to HCV infection, we decided to use a simplified version of the system, since our aim was to focus on the exploration of adaptive mechanisms in hepatocyte-like cells exposed to multiple sources of damage and to investigate the role of individual HCV proteins in this adaptive response.

The increased resistance to oxidative stress, demonstrated by reduced apoptosis, observed in Huh7 cells expressing the HCV proteins Core and NS5A is probably not caused by altered levels of anti-oxidant enzymes. Gene expression of prominent anti-oxidant genes like *SOD1* and *CAT* were not changed by any of the treatments, although we cannot rule out a regulation at the level of antioxidant enzyme activity.

Autophagy has been described as an important adaptive survival mechanism to cope with cellular stress, e.g., ER stress and oxidative stress [26]. The initial step in activation of the autophagic program is activation of the eIF2α/ATF4 pathway [15,27]. In our model, we were able to demonstrate activation of the eIF2α/ATF4 pathway and downstream events like increased ATF4 and CHOP (*DDIT3*) expression. Increased levels of autophagy markers have also been observed in liver biopsies of patients with HCV infection and in cell culture models of HCV infection [28,29]. In our study, we demonstrate increased expression of the transcription factors ATF4 and CHOP (*DDIT3*), key elements of the eIF2α/ATF4 pathway, in liver biopsies from patients with HCV-related cirrhosis, but not in liver tumor tissue of patients with HCV-related HCC or liver tissue of non-HCV-related cirrhosis. These clinical data support our experimental data and also highlight the specificity of the observed changes for HCV during its replication at early stages of the infection and during cirrhosis but not for late stages of the infection as HCC because HCV replication was considered low.

Sir et al. demonstrated that HCV infection induces the accumulation of autophagosomes in cells without increasing autophagic protein degradation, whereas accumulation of autophagosomes and protein degradation were increased in Huh7 cells under starvation, the ‘gold standard’ for induction of autophagy [30]. We did observe increased viral protein degradation in HCV Core and NS5A-expressing Huh7 cells after exposure to external oxidative stress. However, this degradation was independent of the ubiquitin-proteasomal degradation pathway and was also not prevented by blocking autophagic flux with chloroquine. Blocking the lysosomal pathway partially restored HCV protein levels, however additional experiments are required to demonstrate the involvement of selective autophagic pathway and role of autophagy adaptor markers [31].

Microautophagy, in which lysosomes invaginate and directly sequester cytosolic components, has been suggested to play an important role in the elimination of harmful proteins. In this pathway, LC3-II and p62 play an important role as adaptor proteins [32]. In our study, we confirm that selective HCV Core and NS5A degradation can be induced in response to oxidative stress. The identification of autophagy receptor proteins such as p62, which also binds to ubiquitinated proteins, has provided a molecular link between the ubiquitination pathway and autophagy. p62 is a scaffold protein that has been implicated in processes like signal transduction, cell proliferation, cell survival, cell death and oxidative stress response [33] and also plays an important role as receptor for selective autophagy [34]. In our study, the HCV Core and NS5A degradation occurred simultaneously with a decrease of p62 protein level, suggesting the involvement of p62 as receptor protein for degradation of viral proteins after oxidative stress induction. Supporting this hypothesis is our observation that silencing of p62 allows the recovery of the expression of HCV Core and NS5A after external oxidative stress. Since we observed a partial restauration of NS5A expression after p62 silencing (Figure 8D), other mechanism must also be considered. It is also important to demonstrate a direct interaction between HCV Core or NS5A with p62 by immunoprecipitation assay.

Wang et al. suggested that macroautophagy and chaperone-mediated autophagy (CMA) are required for hepatocyte resistance to oxidative stress, because inhibition of macroautophagy sensitized cells to apoptotic and necrotic cell death induced by menadione [35]. Our results indicate a more prominent role for autophagy adaptor proteins like p62. The reason for this difference may be due to differences in the model systems used, but are in accordance with the suggestion of Czaja et al. who propose that two types of autophagy are better than one to face the effects of oxidative stress in hepatocytes and that adaptor proteins play an important role in this effect [26].

In summary, our study demonstrates that hepatocyte-like cells can adapt to multiple stressors, like HCV protein expression and external oxidative stress. We conclude that activation of the eIF2α/ATF4 pathway and subsequent selective degradation, involving LC3-II and p62 contributes to the resistance of hepatocytes to oxidative stress by selective removal of one of the stressors (HCV proteins in this case). This mechanism suggests an important role for autophagy in viral replication and persistence of viral infection and may provide new leads for clinically applicable therapeutic interventions. In addition, it might be suggested to re-investigate in more detail the value of anti-oxidants in HCV patients, since anti-oxidants may abolish the adaptive response of hepatocytes and prevent the degradation of viral proteins in hepatocytes.

## Figures and Tables

**Figure 1 viruses-12-00425-f001:**
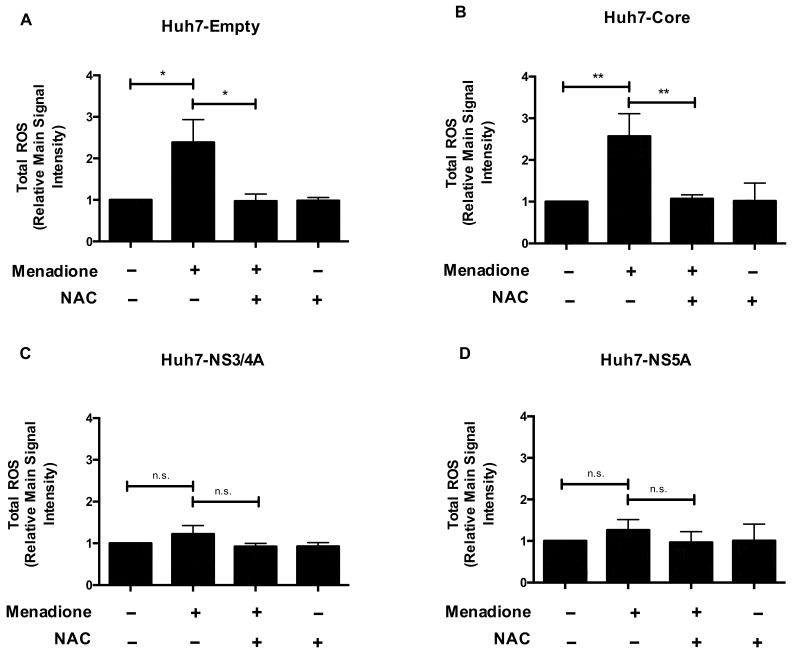
Reactive oxygen species (ROS) generation is attenuated in Huh7 cells expressing NS3/4A and NS5A. Total ROS production was determined in stably transfected Huh7 cells expressing the empty vector (**A**), Core (**B**), NS3/4A (**C**) and NS5A (**D**). Cells were plated in 6-well plates and 24 h post-seeding treated with menadione (50 μmol/L) for 6 h. In some experiments, cells were pre-treated with 5 mmol/L NAC 30 min prior to addition of menadione. ROS production was determined using flow cytometry and the relative main signal intensity is depicted. The graphs show means ± standard deviation (SD) of three independent experiments. A *t*-test was performed to compare the means and the asterisks represent the *p* value ** <0.0093 and * <0.0127. n.s = non-significant, (*p* values > 0.05 are considered not statistically significant).

**Figure 2 viruses-12-00425-f002:**
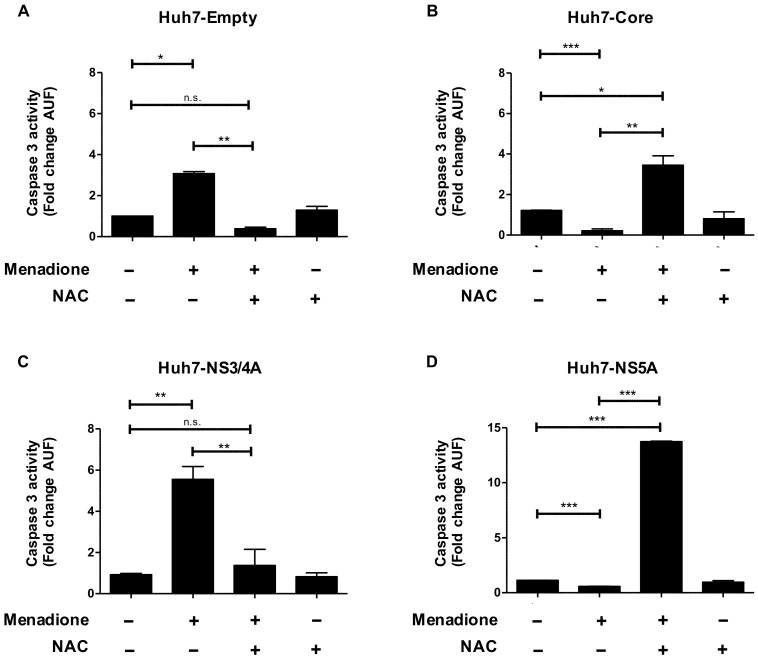
Huh7 cells expressing HCV Core and NS5A are protected against apoptosis induced by oxidative stress. Apoptosis was determined by measuring caspase 3 activity in Huh7 cells expressing (**A**) empty vector, (**B**) Core, (**C**) NS3/4A and (**D**) NS5A after menadione treatment (50 μmol/L). In some experiments, cells were pre-treated with 5 mmol/L *N*-acetyl-l-cysteine (NAC) 30 min prior to addition of menadione. Cells were plated in 6-well plates and 24 h post-seeding treated with menadione for 6 h. Caspase 3 activity was determined as described in Materials and Methods and represented as fold change in arbitrary units of fluorescence (AUF). The graphs depict means ± SD of three independent experiments. A *t*-test was performed to compare the means and the asterisks represent the *p* value *** <0.0009, ** ˂0.002 and * ˂0.02. n.s = non-significant, (*p* values > 0.05 are considered not statistically significant).

**Figure 3 viruses-12-00425-f003:**
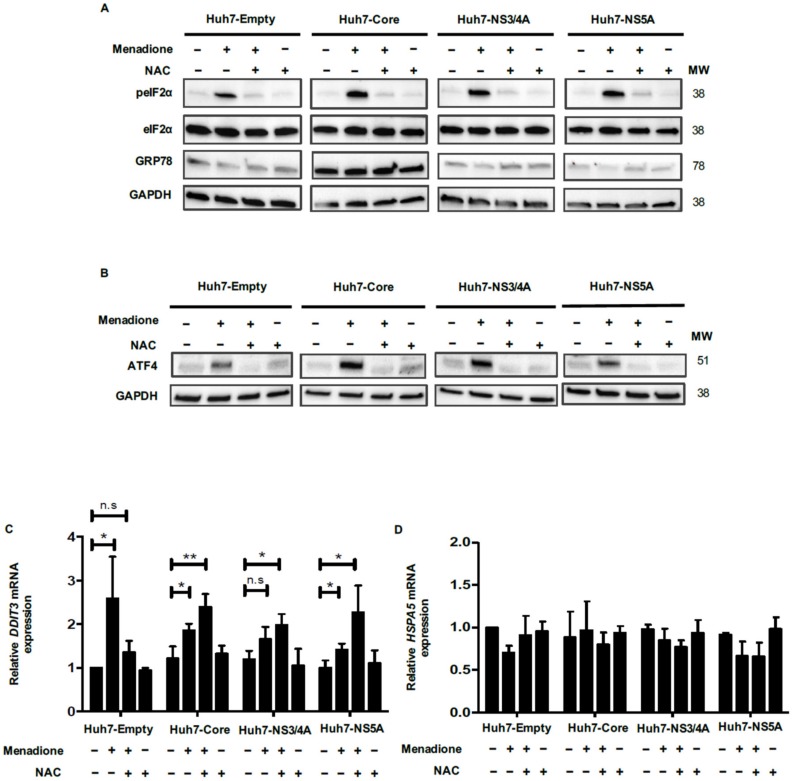
The eIF2α/ATF4 pathway is involved in the adaptive response in the double injury model. Huh7 cells, expressing empty vector, Core, NS3/4A and NS5A HCV proteins were treated with and without 50 µmol/L menadione. In some experiments, cells were pre-treated with 5 mmol/L NAC 30 min prior to addition of menadione. Protein levels of total- and phosphorylated-eIF2α, glucose-regulated orotein of 78kDa (GRP78) and glyceraldehyde 3-phosphate dehydrogenase (GAPDH) (**A**) and ATF4 (**B**) were determined by Western blotting as described in Materials and Methods. mRNA levels of *DDIT3* (**C**) and *HSPA5* (**D**) were evaluated by quantitative polymerase chain reaction (qPCR). The relative expression was normalized based on the expression of 18S. A *t*-test was performed to compare the means and the asterisks represent the *p* value ** ˂0.007 and * ˂0.03. n.s = non-significant, (*p* values > 0.05 are considered not statistically significant).

**Figure 4 viruses-12-00425-f004:**
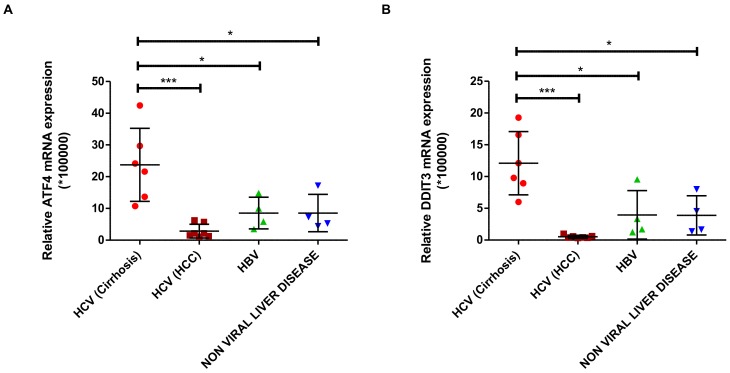
Expression of *ATF4* and *DDIT3* (CHOP) is induced in HCV-related cirrhosis. Total mRNA was isolated from liver tissues obtained from patients with chronic liver diseases and the expression levels of *ATF4* (**A**) and *DDIT3* mRNA (**B**) were determined by qPCR and compared. Patient characteristics are in Table 1. The relative expression of the genes was normalized based on the expression of 18S mRNA. A *t*-test was performed to compare the means and the asterisks represent the *p* value as follows *** ˂0.0006 and * ˂0.04. n.s = non-significant, (*p* values > 0.05 are considered not statistically significant).

**Figure 5 viruses-12-00425-f005:**
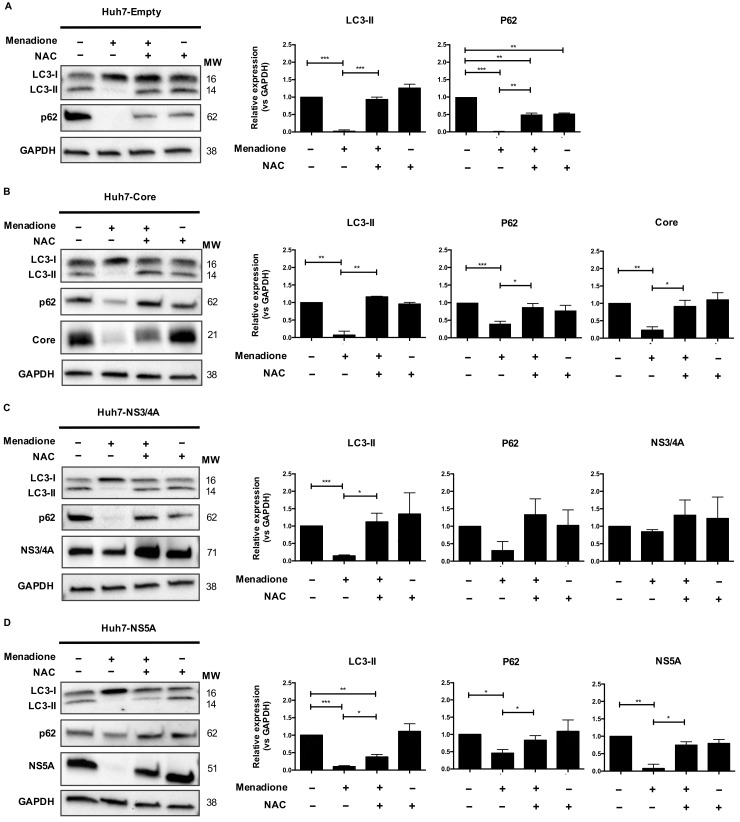
HCV Core and NS5A are preferentially degraded after menadione treatment. LC3-I/II and p62 autophagy markers were detected by Western blotting in Huh7 cells stably expressing the empty vector (**A**), Core (**B**) NS3/4A (**C**) and NS5A (**D**) HCV proteins. Non-treated cells (−/−) were used as control. Oxidative stress was induced by 50 μmol/L menadione (+/−) and in some experiments cells were treated with 5 mmol/L NAC (+/+). Protein band intensities were quantified using ImageLab software (BioRad). The relative protein expression was calculated based on the expression of GAPDH and compared to the expression of the control cells. The graphs depict means ± SD of three independent experiments. A *t*-test was performed to compare the means and the asterisks represent the *p* value *** <0.0007, ** ˂0.0072 and * ˂0.04. (*p* values > 0.05 are considered not statistically significant).

**Figure 6 viruses-12-00425-f006:**
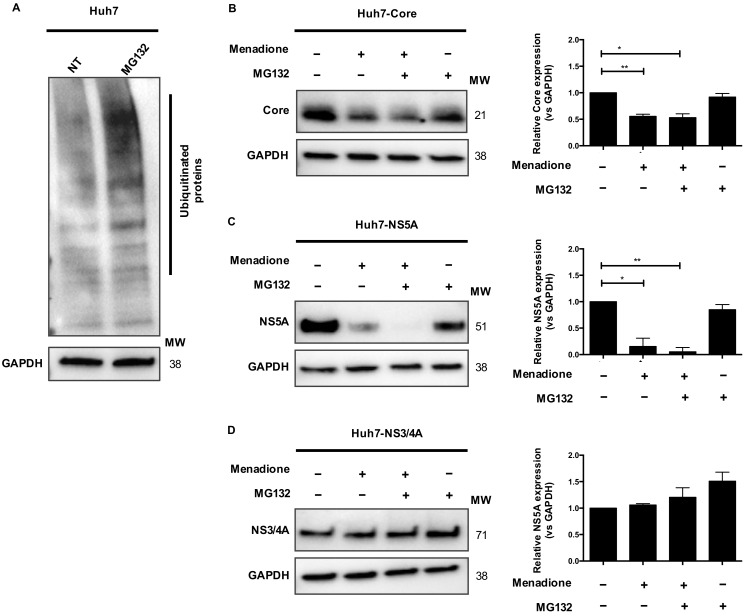
Ubiquitin-proteasome pathway is not involved in HCV Core and NS5A degradation. Huh7 cells were plated in 6-well plates and treated with 10 μmol/L MG132 for 3 h. After treatment, Western blotting was performed to detect ubiquitinated proteins, GAPDH was used as loading control (**A**). Huh7 cells expressing HCV Core, NS3/4A, and NS5A were treated with menadione (50 μmol/L) (+/−). In some experiments 10 μmol/L MG132 was added for 3 h followed by addition of menadione (+/+). The expression of HCV Core (**B**), NS5A (**C**) and NS3/4A (**D**) was determined by Western blotting. GAPDH was used as a loading control. The relative protein expression was calculated based on the expression of GAPDH and compared to the expression of the control cells using ImageLab software (BioRad). The graphs depict means ± SD of three independent experiments. *t* test was performed to compare the means and the asterisks represent the *p* value ** ˂0.0039 and * ˂0.0163. (*p* values > 0.05 are considered not statistically significant).

**Figure 7 viruses-12-00425-f007:**
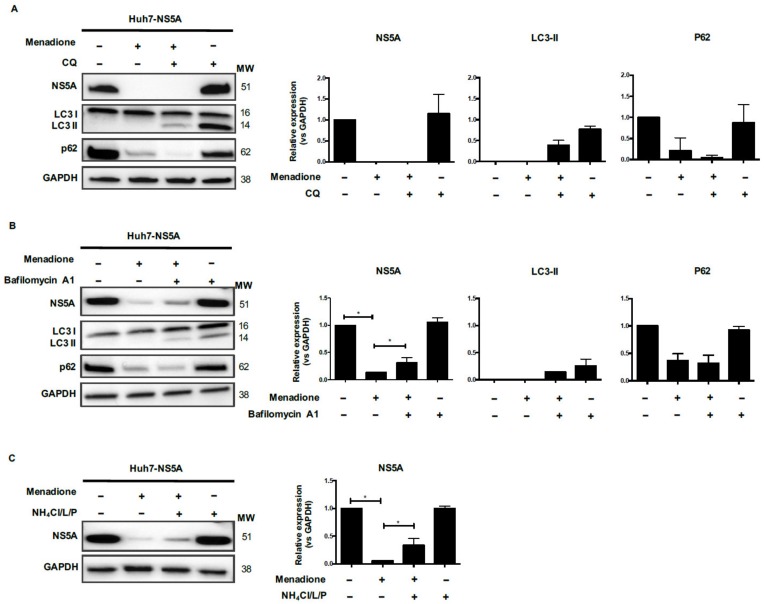
Menadione-induced NS5A degradation involves LC3-II and p62 autophagy markers. Huh7 cells expressing HCV NS5A were exposed or not to 50 μmol/L menadione. Autophagic flux (**A**) was blocked using 50 μmol/L chloroquine (CQ; +/+) and the lysosomal pathway (**B**,**C**) was blocked using bafilomycin A1 (**B**) or NH_4_Cl/leupeptin/pepstatin (**C**). The expression of NS5A, LC3-I/II, p62 and GAPDH were determined by Western blotting. The graphs depict means ± SD of three independent experiments. A *t* test was performed to compare the means and the asterisks represent the *p* value * ˂0.04. (*p* values > 0.05 are considered not statistically significant).

**Figure 8 viruses-12-00425-f008:**
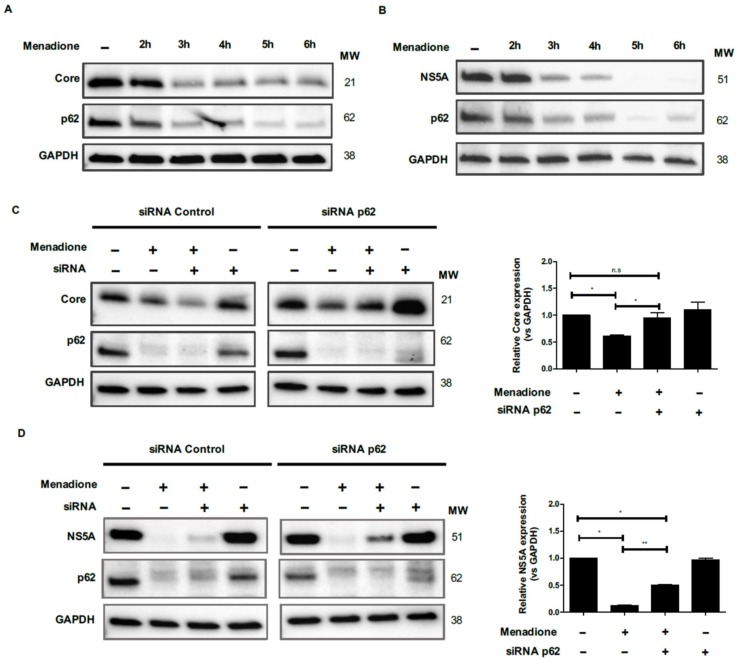
Role for the autophagic adaptor protein p62 in the degradation of Core and NS5A. Huh7 cells expressing Core and NS5A were treated or not with 50 μmol/L menadione. Expression of Core (**A**), NS5A (**B**), p62 and GAPDH proteins were evaluated by Western blotting at 1 h intervals for 6 h as described in Materials and Methods. For p62 silencing, Huh7 cell expressing HCV Core (**C**) and NS5A (**D**) were transfected twice with esiRNAs. Cells were plated and reverse transfected with esiRNA p62 and random esiRNAs as described in Materials and Methods. Transfected cells were treated with menadione (50 μmol/L) and the expression Core, NS5A and p62 protein were detected by Western blot. The experiments were repeated three times. The relative HCV Core and NS5A expression was determined by densitometry and a *t*-test was performed to compare the means and the asterisks represent the *p* value as follows ** ˂0.001 and * ˂0.04. n.s = non-significant, (*p* values > 0.05 are considered not statistically significant).

**Figure 9 viruses-12-00425-f009:**
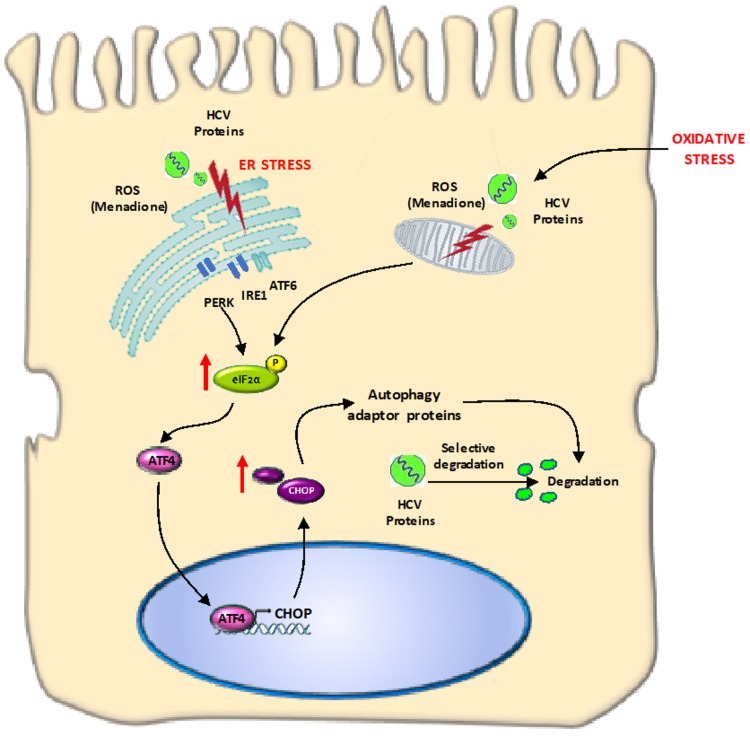
Graphical abstract: HCV proteins are susceptible to degradation after oxidative stress induction. In Huh7 cells expressing HCV viral proteins and under oxidative stress conditions, we observed that HCV Core or/and NS5A proteins were susceptible to degradation after induction of external oxidative stress. The phosphorylation of eIF2α was followed by increased expression of ATF4 and CHOP. Autophagy-related proteins (LC3-II/p62) are involved in the degradation of HCV proteins.

**Table 1 viruses-12-00425-t001:** Liver biopsy samples: characteristics of patients with liver diseases.

Sample No.	Internal Code *	Gender	Age	Infection	Diagnostic
**1**	TH2	M	59	HCV	Cirrhosis
**2**	TH6	M	48	HCV	Cirrhosis
**3**	TH10	F	68	HCV	Cirrhosis
**4**	TH15	F	68	HCV	Cirrhosis
**5**	TH17	F	59	HCV	Cirrhosis
**6**	TH35	M	47	HCV	Cirrhosis
**7**	TH3	M	68	HCV	HCC
**8**	TH7	M	54	HCV	HCC
**9**	TH9	M	53	HCV	HCC
**10**	TH14	M	47	HCV	HCC
**11**	TH16	F	63	HCV	HCC
**12**	TH70	F	61	HCV	HCC
**13**	TH86	M	52	HCV	HCC
**14**	TH33	M	53	HBV	Cirrhosis
**15**	TH42	F	22	HBV	Cirrhosis
**16**	TH43	M	59	HBV	Cirrhosis
**17**	TH75	F	45	HBV	Cirrhosis
**18**	TH19	F	65	-	Cirrhosis
**19**	TH20	F	57	-	Cirrhosis
**20**	TH23	F	57	-	Cirrhosis
**21**	TH28	F	46	-	Cirrhosis

* Sample code at the tissue bank of the Gastrohepatology Group, University of Antioquia. F = Feminine/M = Masculine/- = No viral liver disease.

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
