# Peer review of "Hepatitis C Virus Proteins Core and NS5A Are Highly Sensitive to Oxidative Stress-Induced Degradation after eIF2α/ATF4 Pathway Activation"

_viruses, 2020, doi:10.3390/v12040425_

Round 1

Reviewer 1 Report

The authors answered my questions. 

Reviewer 2 Report

The authors have addressed all concerns I originally raised and I am satisfied with their responses. Congratulations on their interesting work.

This manuscript is a resubmission of an earlier submission. The following is a list of the peer review reports and author responses from that submission.

Round 1

Reviewer 1 Report

Summary: This manuscript focuses on the cellular response to HCV Core and NS5A proteins. The authors used stable cell lines expressing proteins of interest to analyze their effects on oxidative stress, caspase-3 and the eIF2a/ATF4 pathway, as well as autophagy.

Points for Consideration:

Huh-7 cells are not hepatocytes, as written in the abstract. They are a cell line isolated from a liver tumor and should not be referred to as hepatocytes. Perhaps use hepatocyte-like or hepatoma cell line to describe them. The infectious HCV culture system (HCVcc) was discovered in 2005 and is widely used by most researchers in the field to study cellular responses to HCV infection. In this manuscript, the authors have used the much older surrogate model of stable cell lines expressing individual proteins, as many of us did before it was possible to culture the virus. Unfortunately this means the results generated with this system have no physiological relevance since these viral proteins never exist in a cell without ongoing virus replication and the presence of all of the other HCV proteins and replicating genomes. The results in this manuscript need to be validated in an actual viral system. The experiments as performed are in general highly artificial. Oxidative stress is artificially induced by menadione, then it is tested whether single viral proteins are able to protect from NAC abolition of the protective effect? It is well-documented that HCVcc infection of Huh7 cell lines can induce apoptosis, so it appears that the protective effect seen here by single proteins is unique to the artificial situation of single viral protein presence in the cells. To confirm these finding, the authors must validate their work in a HCV infection system. It is nice that the authors attempted to validate their findings in liver samples, but there is a large gap between single protein expression in Huh7 cells and cellular mRNA levels in tissue that had been infected for many years with actual virus. The levels of the studied transcription factors, for example, might indeed be elevated in cirrhotic tissue, but this cannot be attributed to the proteins studied in the stable cells lines. Effects on the host cell observed in the liver tissue could be due to other viral proteins not studied here, or the general antiviral, stress-inducing, inflammatory environment that is present in chronically infected cirrhotic liver.

Reviewer 2 Report

Viral infection is intimately associated with the stress response machinery of the host. Here Riso-Ocampo, et al. examined the regulation of hepatitis C virus (HCV) proteins core and NS5A in response to oxidative stress. They found that HCV NS3/4A and NS5A attenuate ROS generation and HCV Core and NS5A suppress apoptosis induced by oxidative stress. The further demonstrated that HCV core and NS5A are selected degraded by menadione treatment independent of ubiquitin-proteasome pathway and autophagy pathway was partially involved in the degradation of core and NS5A.

Specific comments,

Fig 2B and 2D, It is counterintuitive that menadione treatment even decrease caspase activation compared with non-treatment control. Are there any explanations? Additionally, western blot showing cleaved caspase-3 should be done to corroborate the results.

Fig 3C, could the fold increase reach statistical significance (Huh7-Empty) if the experiments are repeated multiple times. If this is the case, then HCV protein expression does not affect DDIT3 gene expression.

Fig 7, the authors should check the mRNA level to determine the protein vs RNA regulation by menadione. The individual inhibitors barely rescue the protein expression. The authors should also examine whether the combination of inhibitors (eg. MG132, CQ, NH4Cl etc) could restore the protein expression.